# First Year of TREC-Based National SCID Screening in Sweden

**DOI:** 10.3390/ijns7030059

**Published:** 2021-08-25

**Authors:** Christina Göngrich, Olov Ekwall, Mikael Sundin, Nicholas Brodszki, Anders Fasth, Per Marits, Sam Dysting, Susanne Jonsson, Michela Barbaro, Anna Wedell, Ulrika von Döbeln, Rolf H. Zetterström

**Affiliations:** 1Centre for Inherited Metabolic Diseases, Karolinska University Hospital, 17176 Stockholm, Sweden; sam.dysting@sll.se (S.D.); susanne.v.jonsson@sll.se (S.J.); michela.barbaro@sll.se (M.B.); anna.wedell@ki.se (A.W.); ulrika.vondobeln@sll.se (U.v.D.); 2Department of Molecular Medicine and Surgery, Karolinska Institutet, 17176 Stockholm, Sweden; 3Department of Pediatrics, Institute of Clinical Sciences, The Sahlgrenska Academy at University of Gothenburg, 40530 Gothenburg, Sweden; olov.ekwall@gu.se (O.E.); anders.fasth@gu.se (A.F.); 4Department of Rheumatology and Inflammation Research, The Sahlgrenska Academy at University of Gothenburg, 40530 Gothenburg, Sweden; 5Department of Clinical Science, Intervention and Technology, Karolinska Institutet, 17177 Stockholm, Sweden; mikael.sundin@ki.se (M.S.); per.marits@sll.se (P.M.); 6Section of Pediatric Hematology, Immunology and HCT, Astrid Lindgren Children’s Hospital, Karolinska University Hospital, 14186 Stockholm, Sweden; 7Department of Pediatric Immunology, Children’s Hospital, Lund University Hospital, 22242 Lund, Sweden; nicholas.brodszki@skane.se; 8Department of Clinical Immunology and Transfusion Medicine, Karolinska University Hospital, 14186 Stockholm, Sweden; 9Department of Medical Biochemistry and Biophysics, Division of Molecular Metabolism, Karolinska Institutet, 17177 Stockholm, Sweden

**Keywords:** SCID, neonatal dried blood spot screening, TREC, KREC, sensitivity, specificity, PPV, NPV

## Abstract

Screening for severe combined immunodeficiency (SCID) was introduced into the Swedish newborn screening program in August 2019 and here we report the results of the first year. T cell receptor excision circles (TRECs), kappa-deleting element excision circles (KRECs), and actin beta (ACTB) levels were quantitated by multiplex qPCR from dried blood spots (DBS) of 115,786 newborns and children up to two years of age, as an approximation of the number of recently formed T and B cells and sample quality, respectively. Based on low TREC levels, 73 children were referred for clinical assessment which led to the diagnosis of T cell lymphopenia in 21 children. Of these, three were diagnosed with SCID. The screening performance for SCID as the outcome was sensitivity 100%, specificity 99.94%, positive predictive value (PPV) 4.11%, and negative predictive value (NPV) 100%. For the outcome T cell lymphopenia, PPV was 28.77%, and specificity was 99.95%. Based on the first year of screening, the incidence of SCID in the Swedish population was estimated to be 1:38,500 newborns.

## 1. Introduction

Severe combined immunodeficiency (SCID) describes a group of molecularly diverse diseases characterised by malfunction of the adaptive immune response due to the absence of T lymphocytes, B lymphocyte dysfunction or absence, and in certain conditions, also the absence of natural killer cells [1]. According to the latest report of the International Union of Immunological Societies (IUIS, Berlin, Germany) defects in 18 genes are recognised to cause SCID [2].

Children affected by SCID appear healthy at birth in most cases but often present within the first six months of age with intractable diarrhoea, failure to thrive, pneumonia, and recurrent, treatment-resistant infections, frequently caused by opportunistic pathogens. In addition, live vaccines such as rotavirus and BCG vaccines, which are included in many vaccination programs, can be life-threatening to a child with undiagnosed SCID. Untreated, the condition is fatal and children with SCID usually die within the first two years of life [1,3,4,5].

Treatment today consists of either hematopoietic cell transplantation (HCT), enzyme-replacement therapy (ERT), or gene therapy depending on the underlying molecular defect and HCT donor availability [3]. Therapeutic success, however, as assessed by 2-year survival, is largely dependent on the absence of active infections. Survival rates following HCT have been reported to be 95% if the transplantation was performed in children without a prior history of infection but dropped to 81% in children with an active infection at the time of transplantation, thus necessitating the detection of SCID cases in the early neonatal period to minimise the risk of infection [6].

An assay for neonatal SCID screening from Guthrie cards (dried blood spots (DBS)) was developed in 2005 by Chan et al. [7]. The qPCR-based assay approximates thymic output by quantitation of δRec-φJα TRECs which are produced by 70% of developing human αβ T cell receptor (TCR) expressing T lymphocytes as a by-product of the V(D)J recombination of their TCR gene [8,9,10]. While the sensitivity of the screening test has been reported to be 100%, the PPV and the specificity vary depending on the definition of true positives (T cell lymphopenia or SCID) and the cut-off values [11,12].

SCID screening was first introduced in Wisconsin (USA) in 2008 [13] and included in the Recommended Uniform Screening Panel in the USA in 2010 [14]. In Europe, national SCID screening was first included in the newborn screening program of Iceland in 2017 (personal communication with Una Bjarnadottir), followed by Norway in 2018 [15], and it has now been implemented in several other European regions and countries [16].

In Sweden a pilot study was carried out in the County of Stockholm (Region Stockholm, Sweden) between 15 November 2013 and 14 November 2016, analysing TREC and KREC levels in 89,462 children, allowing thus the detection of both T and B lymphocyte deficiencies. The incidence of SCID was estimated to be approximately 1:45,000 births in Stockholm County [17,18,19].

We report here the results of the first year of the Swedish national screening program for SCID using a commercially available kit detecting TREC, KREC, and ACTB levels. 

## 2. Materials and Methods

### 2.1. Samples

As part of the newborn DBS screening in Sweden, venous blood samples were collected on PerkinElmer 226 Ahlstrom filter paper (Guthrie card), in four spots, as soon as possible after 48 h of age from a vein at the back of the hand of all newborns, whose parents opted to participate (>99.5% of newborns). In addition to newborns, samples from children immigrating to Sweden, who were between one month and two years old, were included in the screening for SCID. The DBS samples were air-dried and sent to the laboratory by regular mail. All screening cards of the cohort (115,786 samples) included in this report arrived in the laboratory between 5 August 2019 and 4 August 2020.

### 2.2. Screening Assay

TREC, KREC, and ACTB levels were analysed in 3.2 mm punches of the original screening card according to the manufacturer’s instructions using the SPOTit-TK kit in 96-well format (ImmunoIVD, Nacka, Sweden). DNA elution was performed in MiniAmp Thermal Cyclers (Thermo Fisher Scientific, Waltham, MA, USA), and the real-time PCR analysis was carried out on Applied Biosystems QuantStudio 5 Dx instruments (Thermo Fisher Scientific, Waltham, MA, USA) using the QuantStudio 5 Dx IVD Software v1.0.

On each 96-well plate, three internal DBS controls (low-TREC/high-KREC, low-KREC/high-TREC, and low-TREC/low-KREC), and one blank PCR (no DBS) were analysed. Quantification was carried out using 5-point standard curves (provided in the kit) ranging from 10 to 100,000 copies/well, for TREC, and KREC and from 100 to 1,000,000 copies/well for ACTB. 

Results of a plate were accepted if the standard curve, as well as the internal controls, fulfilled the requirements described in the kit protocol.

Analytical limits of the assay, reported by the manufacturer, were as follows: limit of detection for TREC 3.41 and for KREC 3.13 copies/well; limit of blank for TREC 0.32 and for KREC 0.38 copies/well.

### 2.3. Screening Algorithm

Amplification plots were visually inspected for all samples in the QuantStudio 5 DX software prior to data export. Sample quality was considered sufficient if all amplification curves showed the expected amplification profile and ACTB was ≥1000 copies/well. 

Samples were considered directly screening negative if the TREC concentration was above the reanalysis cut-off 15 copies/well (lowered to TREC ≤ 10 copies/well, on 1 April 2020).

If the TREC result was between the reanalysis and the referral cut-off, samples were reanalysed in duplicate from different blood spots of the original screening card. If the TREC result was below the referral cut-off, samples were analysed in quadruplicate, or from all blood spots in cases where fewer than four spots were available.

The referral cut-off was TREC ≤ 6 copies/well throughout the reporting time, according to the manufacturer’s recommendation. If all the replicate analyses yielded TREC levels below the referral cut-off, and ACTB levels were simultaneously below 1000 copies/well, samples were considered inconclusive due to amplification failure, and a new DBS card was requested (Figure 1).

Samples were considered screening positive if the mean TREC concentration of all replicates’ PCRs was ≤6 copies/well. In case of extreme variability in the PCR results of a sample, individual replicates with particularly low TREC results were removed from the calculation and from the decision to refer the child to the specialist care centre, since we argued that samples from SCID cases should yield uniformly low TREC results. Borderline cases were referred at the discretion of the clinician in charge.

KREC levels were not included in the decision to refer children for further investigation as decided by the Swedish National Board of Health and Welfare. This decision was based on the uncertainty of whether X-linked agammaglobulinemia (Bruton’s disease) qualifies for NBS and a high false-positive rate due to KREC-based referrals in our pilot study [18]. However, for screening positive children the KREC levels were reported alongside TREC values to paediatric immunologists at one of three tertiary paediatric centres in Sweden (Skåne University Hospital in Lund, Sahlgrenska University Hospital in Gothenburg, and Karolinska University Hospital in Stockholm), who contacted the families and performed or organised clinical investigations and diagnostics. Follow-up screening cards were obtained at the visit to the specialist centre or at the local hospital and mailed to the laboratory for analysis.

The primary outcome for the screening was the detection of SCID, the secondary outcome was T cell lymphopenia. False positives (FPs) for the primary outcome were all children referred to specialist care centres who were not diagnosed with SCID; FPs for the secondary outcome were children who showed no clinical sign of lymphopenia. 

### 2.4. Clinical Procedures

Referred children were clinically examined, and total lymphocyte count, as well as FACS analysis, was performed to determine lymphocyte subsets according to established procedures [20]. Children not living in the Gothenburg, Lund, or Stockholm regions or premature children sampled at neonatal units were examined at local hospitals or by neonatologists with advice from specialists from one of the three centres.

For all SCID cases and for cases with severe T cell lymphopenia, whole-genome sequencing was carried out at SciLifeLab Stockholm through the Genomics Medicine Centre Karolinska (GMCK, Solna, Sweden), and results were interpreted at the Department for Clinical Immunology (Karolinska University Hospital, Solna, Sweden) [21]. Lymphopenia was defined as CD3^+^ T cells below 2 × 10^9^ cells/L [22].

### 2.5. Quality Control

For internal quality control, plate medians, minima, maxima, 5th and 95th percentiles were recorded for all analytes and compared across time and kit lots. The ability of the different kit lots to detect SCID patients was assessed using DBS from adult controls, who had TREC concentrations below the referral cut-off. In addition, the laboratory participates in the CDC proficiency testing program (TRECPT). 

### 2.6. Data Curation, Analysis, and Statistics

Demographic information from the screening cards was scanned and stored in the Labware 7 lab information system (Labware Inc., Wilmington, DE, USA). PCR results were exported from the instrument software to Excel for further analysis.

Data analysis and visualisation were performed using the R package tidyverse 1.3.0 within R-Studio version 1.1.456 [23]. Gaussian distribution of values was assessed using Shapiro–Wilk test for normality where applicable. 

In addition, 95% confidence intervals (CI) for proportions were estimated using Wilson’s score in the R package PropCIs 0.3–0.95, and confidence intervals for the incidence calculation were calculated using the exact method of the epiR 1.0–15 package. To analyse the TREC distribution in the screened population, samples with unsatisfactory TREC qPCR amplification profiles or samples in which ACTB was below 1000 copies/well, and TREC below the reanalysis cut-off were removed from the dataset. The remaining results were averaged to obtain one result per screening card. The distribution of KREC values was obtained from the same samples.

## 3. Results

### 3.1. Demographics

During the reporting time, samples from 115,786 children up to 2 years of age arrived at the laboratory of which 55,787 (48.18%) were female, 59,985 (51.81%) were male, and for 14 the sex was not noted in the database. The dataset contained 115,216 newborns and 570 children between 28 days and 2 years of age (non-newborns). Among the newborns, 108,524 were term babies (born ≥ week 37), 5704 were moderate preterm (week 32–36), 642 very preterm (week 28–31), 341 extremely preterm (<week 28), and for 5 infants gestational age was unknown (Figure 2a). The median age at sampling was 58.1 h (IQR 49.9, 73.9) for term babies, and 57.1 h (IQR 49.9, 73.0) for the entire population of newborns included in this study (Figure 2b). Approximately 65% of all newborn samples were collected between 48 h and 72 h of age, which is in accordance with the sampling instructions.

### 3.2. TREC and KREC Results

To analyse the distribution of TREC levels in the screened population, the mean per infant was calculated based on PCR results that met acceptance criteria. The TREC levels of the population ranged from 0 to 5419 copies/well, 95% of samples had a TREC concentration below 171 copies/well. The population median was 79 TREC copies/well (IQR 55, 110) (Figure 3a). For the same samples, the distribution of KREC values was plotted (Figure 3b). KRECs ranged from 0 to 2561, with 95% of samples falling below 126 copies/well and a population median of 51 copies/well (IQR 34, 75). Compared to TRECs, the distribution of KRECs was much narrower.

We further compared TREC and KREC values between newborns delivered at different gestational ages. To this end, the screened newborn population was subdivided into blocks of 3 weeks starting at gestational week 22, and for each group, density plots were generated. For five newborns, information on gestational age was missing. Comparison between the groups showed that the peaks of the TREC distributions shifted gradually towards higher values with increasing gestational age at birth, indicating an increase of the frequency of higher TREC values in more mature newborns (Figure 3c, Table 1). 

The peaks of the KREC distributions, on the other hand, did not show any shift with increasing gestational age, and moreover, the shape of the distributions was very similar. This observation suggests that there was no difference in KREC values in very prematurely born, compared to term-born infants (Figure 3d, Table 1). 

Non-newborns up to the age of two years are included in the Swedish SCID screening. Their TREC and KREC values were analysed in more detail and compared to the values obtained from term newborns. The median TREC concentration in non-newborns was 104 copies/well (IQR 64, 151) and 80 copies/well (IQR 56, 110) in term newborns, showing only a small shift in the median value and largely overlapping interquartile ranges. Density plots of the TREC values showed that the spread of the distribution was wider for non-newborns, compared to term newborns, with a higher frequency of both higher and lower TREC values in the older children (Figure 3e). The KREC distribution of non-newborns showed a wider spread in conjunction with a right shift, compared to KREC results in term newborns. The median concentration was 180 copies/well (IQR 126, 258) in non-newborns and 51 copies/well (IQR 34, 74) in term newborns leading to a separation of the peaks of the density plots (Figure 3f).

Grouping non-newborn children by age did not reveal any trends for TRECs but showed rather stable levels during the first 14 months of life with a slight decrease later on. KRECs increased slightly in the early postnatal phase only to decrease after about five to six months of age. However, the spread of the results was large and the number of samples per group was small (data not shown).

During the reporting period, 1428 samples needed reanalysing because of amplification profiles that did not meet acceptance criteria or because of TREC levels below the reanalysis cut-off. During the first eight months of screening, until 31 March 2020 (period 1), the TREC reanalysis cut-off was 15 copies/well, and the reanalysis rate was 1.55%. Closer examination of the reanalysed samples revealed that for all samples, for which the TREC result of the first PCR analysis was above 8.7 copies/well, the mean TREC value remained above the referral cut-off (data not shown). Therefore, we were confident to lower the reanalysis cut-off to 10 TREC copies/well. This change resulted in a decrease in the reanalysis rate, reducing to 0.68% between 1 April 2020 and 4 August 2020 (period 2). For the entire year, the reanalysis rate was 1.23%. For 27 children (20 newborns, 7 non-newborns), no valid PCR result could be obtained from the original Guthrie card, despite the analysis of punches from all available blood spots, and second DBS samples were requested due to inconclusive sample quality. We obtained repeat samples from 20 children, who were screening negative upon analysis of the second DBS card, and seven children were lost to follow-up (four newborns, three non-newborns). The repeat rate, due to inconclusive sample quality, during the first year of screening was 0.023%. Upon investigation, we found that for several of the inconclusive newborn samples, coated glass capillaries had been utilised for sample collection, while some of the non-newborn samples were most likely collected using tubes with additives.

### 3.3. Referrals

During the first year of SCID screening, 73 children with mean TRECs below 6 copies/well (referral rate = 0.063%) were referred for clinical investigation. Of those, 35 were preterm children (referral rate = 0.52%), 33 were term children (referral rate = 0.03%), and five were non-newborns (referral rate = 0.87%). The rate of samples with undetectable TRECs was 0.016% (TREC < 1 copies/well, *n* = 19, of which 16 were newborns, 10 were term, and 6 were preterm children).

The five referred non-newborn children were sampled at a median age of 532.5 days (IQR 464.5, 634) and visited the specialist centre at a median age of 644.5 days (IQR 541.6, 694.4). Three of those children had undetectable TREC levels on their first screening card; however, for all five children, SCID could be ruled out clinically, and the TREC values were found to be normal in the new DBS sample. 

For referred newborns (*n* = 68), the median age at sampling was 53.5 h (IQR 49.9, 61.9), the median age at referral to the specialist centre was 7.9 d (IQR 6.4, 8.9), and the family was contacted by a clinician either the same or the following day. Further diagnostic testing and the sampling of the second screening card was carried out at a median age of 9.3 days (IQR 8.4, 11.6) for term newborns (*n* = 28) and at 47.8 days (IQR 17.25, 97.2) for preterm children (*n* = 27). In total, 13 children were excluded from these calculations: for two children, the information regarding the sampling date of the second screening card was incomplete, for three children the second screening card had been routinely sampled before the referral because they had received TPN treatment and were therefore not included in the calculations, five prematurely born children had died for reasons unrelated to SCID, and for three newborns, no follow up DBS sample was taken.

In total, 21 referred children, all of them newborns, were true positives (TP) (PPV_lymphopenia_ = 28.77%, Table 2) for the outcome T cell lymphopenia. Three of them were diagnosed with typical SCID (PPV_SCID_ = 4.11%, Table 2). Based on these numbers the SCID incidence was estimated to be 1:38,500 newborns (95% CI: 1:200,000–1:13,200). Molecularly, two of the SCID cases were caused by mutations in the *JAK3* gene, and one case was caused by mutations in the *ADA* gene. The child diagnosed with ADA-SCID was started on ERT from the age of 35 days and was subsequently treated with genetically corrected autologous stem cells at the age of 224 days. HCT in the two JAK3-SCID cases was performed at the age of 56, respectively, 78 days. At the time of reporting, all three children were alive and well. 

Of the remaining eighteen infants with T cell lymphopenia, eight were diagnosed with genetic syndromes (four with 22q11 deletion syndrome (22q11DS), one with CHARGE syndrome, four with other syndromes), two children suffered from chylothorax, one from hydrops, one from postnatal sepsis, and one from juvenile myelomonocytic leukaemia. For four children, the reason for the lymphopenia could not be clearly defined, but prematurity was most likely the reason in one case (Table 3). Inversely, T cell lymphopenia was diagnosed in only 11 of the 19 children who had undetectable TRECs. Inexplicably, the follow-up screening card of one of the children with idiopathic lymphopenia showed very high TREC levels (180 copies/well, *n* = 3 punches, similar result upon repeat with another kit lot), while clinical examinations performed at the same time showed a clear lymphopenia (0.33 × 10^9^ CD3^+^ cells/L).

The 52 false-positive cases included 6 children with genetic syndromes (2 with 22q11DS, 4 with other syndromes), 1 child each with hydrops or chylothorax, 1 child whose mother had been treated with immunosuppressants, and 1 older child with Hepatitis A and B infection. While prematurity could have been the reason for the low TREC levels at the time of initial sampling in 23 newborns, no explanation could be found for the remaining 19 of the FPs.

Until the time of reporting (follow up time between 11 and 23 months), we have not been informed of any undetected SCID cases among the children screened during the first year (NPV_SCID_ = 100%, Table 1). We were informed, however, of the birth of one child with combined immunodeficiency (MHC class II deficiency) who was not detected by the screening. Reanalysis of the child’s primary screening card confirmed TREC values above cut-off (35 copies/well, mean of three analyses), while TREC values at eight months of age had decreased to 3 copies/well (mean of three analyses).

### 3.4. Post Hoc Analysis of the Screening Algorithm

The goal of newborn screening for SCID is primarily the detection of classical patients with SCID who die before the age of two years if undiagnosed and who have a significantly worse outcome if diagnosed late. As mentioned above, the PPV of our current screening is 4% for the outcome SCID and 29% for lymphopenia (PPV_lymphopenia_ for preterms was 20% (95% CI 10–35.9), for term infants 42% (95% CI 27.2–59.2), and there were no preterm SCID cases in our cohort).

Aiming at lowering the number of false-positive cases through a refinement of the referral procedure without impacting sensitivity, we analysed the group of screening positive children and their TREC results in more detail. Specifically, we investigated if lowering the referral cut-off was possible without decreasing sensitivity and if a special referral procedure for preterm children would be adequate.

To explore the possibility of lowering the referral cut-off, the TREC values of referred children were plotted classified by the clinical diagnosis of the child (Figure 4a).

This graph revealed that while the TREC values of false positives were spread out over the entire referral interval, TPs tended, with few exceptions, to accumulate in the lower half. Most importantly, TRECs in all three SCID cases were below 1 copy/well with our method. Statistical analysis comparing TREC values between patients with SCID, patients with T cell lymphopenia and FP confirmed a significant difference in TRECs between patients with SCID and FP, while there was no difference when comparing T cell lymphopenic children to FP or SCID cases (ANOVA, df = 2, F = 5.906, *p* = 0.00363; Bonferroni post hoc test, *p*_SCID//FP_ = 0.022).

Lowering the referral cut-off to ≤4.0 TREC copies/well in a post hoc analysis decreased the number of referred children to 53. At this cut-off two T cell lymphopenic newborns suffering from genetic syndromes, 1 22q11DS child and 17 FPs would not be referred for further examination (Figure 4a). The PPV_SCID_ would therefore increase to 5.6% (95% CI 1.9–15.4), and the PPV_lymphopenia_ to 34.0% (95% CI 22.7–47.4), while sensitivity_SCID_ would remain unchanged. 

TREC levels in preterm children were on average lower than in children born at term (Figure 3b), and their referral rate was 17 times the referral rate of term newborns (0.52% vs. 0.03%). Analysis of the proportions of newborns of different gestational ages within the group of referred children, compared to their proportions in the screened population (Figure 4b), showed that while 93.7% of the screened newborns were born at term, they represented only 48.5% of the referred newborns. The exact composition of the group of referred children is shown in Figure 4b. Due to this imbalance, we investigated how a uniform change in referral cut-off would affect the number of referred term and preterm children: At a referral cut-off of 4 TREC copies/well, 23 preterm (6 TP) and 25 term (12 TP) children would be included in the referrals, resulting in an increase of the PPV_lymphopenia_ for preterm children from 20% to 26% (95% CI 12.6–46.5), and for term children from 42% to 48% (95% CI 30.0–66.5), compared to the cut-off in use. 

Since newborns born preterm do not have a higher a priori risk for SCID than term children but on average lower TREC levels, further adjustments to the referral procedure were investigated with the aim to reach similar referral rates and PPV between the groups. At the hypothetical cut-off of 4 TREC copies/well, 0.023% (~1:4350) of term children would be referred. The equivalent percentile within the preterm population would result in exclusively referring children with 0 TREC copies/well. This finding, together with the limit of detection of the method of 3.41 copies/well, suggests that equal referral rates in term and preterm children are impossible to achieve using only the TREC assay in the screening procedure.

## 4. Discussion

Since the incorporation of SCID newborn screening into the Wisconsin program in 2008, SCID newborn screening has been included in multiple screening programs around the world [16,24]. All screening programs use PCR-based proprietary or commercially available methods to detect TRECs as a proxy for thymic output, with few programs also referring children based on low KRECs.

In Sweden, a regional pilot study was carried out between 2013 and 2016 in which two patients with SCID and three patients with combined immunodeficiency (CID) were detected. In this study, children were referred for further examination based on low TRECs and or low KRECs [17,18]. Despite the positive results of the study and a positive decision from the screening council at the Swedish Board of Health and Welfare, SCID screening was introduced at the national level as late as August 2019 subsequent to a change in the Swedish biobank law allowing storing of cards after screening for other disorders than congenital hypothyroidism and inborn errors of metabolism. 

The vast majority of children that were screened for SCID in Sweden between 5 August 2019 and 4 August 2020 were born full term (93.7%), and 5.8% were preterm newborns, placing Sweden at the lower end of the preterm birth rate in Europe and worldwide [25]. Only 0.5% of children in our cohort were older than one month at the time of analysis.

The TREC and KREC values presented here were generated using a commercial real-time PCR kit which quantifies patient samples relative to a plasmid-derived standard curve. Median TREC and KREC values obtained in the Swedish population were similar to results obtained with the same assay in a prospective screening study carried out in the Polish-German border region using the same kit [26].

Our analysis of the TREC results by gestational age showed that within the newborn group, TREC distributions shifted to higher TREC levels with increasing gestational age at birth, consistent with a rise in TREC medians and in agreement with previously published data [17,26,27,28,29]. The scientific literature is currently less clear on the influence of gestational age on KREC levels, with studies reporting stable KREC levels, independent of gestational age [26,30], as well as increasing KRECs over time, similar to what has been reported for TRECs [17,29]. Our analysis of KRECs showed highly similar distributions of KREC concentrations in newborns of different gestational ages, lending further support to the hypothesis that KRECs in newborns are independent of the gestational age at birth. The current consensus on the maturation of the immune system is, however, that both T and B lymphocyte counts (which are indirectly measured by TRECs and KRECs) are higher in term than in preterm newborns [31,32]. Although the TREC results agree with these findings, more work is needed to reconcile the data on KRECs and B lymphocyte counts. 

To our knowledge, this is the first study reporting TREC and KREC values from screened non-newborn children. To gain further insight into the postnatal TREC and KREC levels, we compared them to those in term newborns and analysed their levels at different postnatal ages. By and large, TRECs did not differ much between term newborns and children above one month of age, while KREC values showed a much wider distribution and were on average higher in older children.

During the first year of screening, 73 children with low TREC values were referred for clinical examination, resulting in a referral rate of 0.063%. Other screening programs have reported referral rates between 0.02% and 0.17% using either proprietary methods or commercial systems from different manufacturers [11,15,26,28,33,34,35,36]. The comparatively high referral rate in our study can be explained by the fact that only one referral cut-off was used leading to contact with a specialist care centre without delay, independent of gestational age. Other programs requested a second screening card for result verification in cases of TRECs below cut-off but detectable or for premature children with abnormal results [15,28,34,35,36,37].

Similar to our study, PPV*_SCID_* in other reports using the TREC assay without a second-tier method was consistently below 10% [28,35,36,37,38]. Gizewska et al. obtained a TREC-based PPV*_SCID_* of approximately 17% with a two-step referral procedure, according to which requesting a second screening card due to low but detectable TRECs was not counted as a referral [26].

Using 6 TREC copies/well as cut-off, we detected, to our knowledge, all SCID cases born in Sweden during the reporting period. We also detected 18 children with T cell lymphopenia, of which 1 child was diagnosed with juvenile myelomonocytic leukaemia at the age of 10 days, 2 days after the second screening card was sampled. Those children profited to a varying degree from the referral to paediatric immunologists for further examination even if they were not the primary target of the screening as mandated by the National Board of Health and Welfare. An additional 52 FPs who had no benefit from the ensuing clinical examination were part of the referred group. 

It is worth noting that one child with MHC II deficiency was detected clinically during the study period. This severe immunodeficiency disorder is commonly missed by the TREC assay since T lymphocytes are present albeit non-functional [39,40]. The screening test in this child was far above our rerun level but interestingly, a new test performed at eight months of age showed only 3 TREC copies/well. 

Even though all cases of SCID that were detected with the assay we were using have been reported to have TREC levels below 1 copy/well (based on this study and [26,41]), there is substantial overlap in TREC results between SCID cases, T cell lymphopenia cases, and even FP. TREC levels of the 52 FP in our study ranged from 0 to 6.4 copies/well, TREC levels in children diagnosed with T cell lymphopenia from 0 to 4.9 copies/well. Importantly, for 19 FP, no clinical reason for TREC levels below the referral cut-off could be determined. Taken together these findings demonstrate the possibility to lower the referral cut-off, and at the same time, highlight the need for the second-tier approaches for substantial improvement of the referral rate and the PPV without compromising sensitivity. The Norwegian SCID screening program confirmed disease status genetically by whole-exome sequencing and analysis of disease-relevant genes before referring children with abnormal but detectable TRECs and obtained a recall rate of 0.02%. This figure excludes, however, preterm children and children in NICU units with intermediate TREC levels for whom no mutations could be found in the NGS step. Those were considered screening negative [15]. A recent study carried out in the Netherlands explored epigenetic immune cell counting and a second TREC assay as non-genetic second-tier approaches to reduce the number of false-positive referrals [42].

In total, five cases of SCID have been identified by newborn screening in Sweden since 2013 (2013–2016 and 2019/2020). Genetically, we found two cases *JAK3* SCID, two *ADA* SCID, and one case *DCLRE1C* SCID (based on this study and [18]). While the case numbers are low, it should be noted that none of the cases detected in Sweden by newborn screening carries a mutation in the *ILR2G* gene, which is by far the most frequent mutation in North America [43]. Moreover, between 2016 and 2019, no SCID cases caused by mutations in the *ILR2G* gene have been identified clinically.

Based on the first year of screening the incidence of SCID in Sweden was estimated to be 1:38 500 newborns with a very wide 95% confidence interval due to the low number of detected cases. A combination of the dataset from Zetterström et al. [18] with the results from the present study allows to estimate the incidence in Sweden with a narrower confidence interval to 1:42,700 newborns (95% CI 1:125,000–1:17,500), based on approximately 205,000 screened children. The incidence in Sweden is in good agreement with the incidence rates of 1:58,000 reported for the USA and of 1:63,000 for France [11,44] but lower than the SCID incidence in Israel (1:22,000) [27] and higher than in Taiwan (1:131,000) [33].

## Figures and Tables

**Figure 1 IJNS-07-00059-f001:**
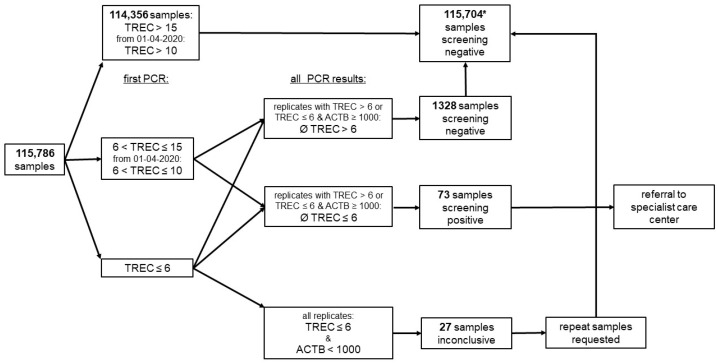
Screening algorithm. Analyte concentrations are reported in copies/well, and the referral cut-off was TREC ≤ 6 copies/well; * 2 parents refused newborn screening but sent screening cards to the laboratory, 7 children with inconclusive samples were lost to follow-up, and 20 had normal values upon testing of a new screening card.

**Figure 2 IJNS-07-00059-f002:**
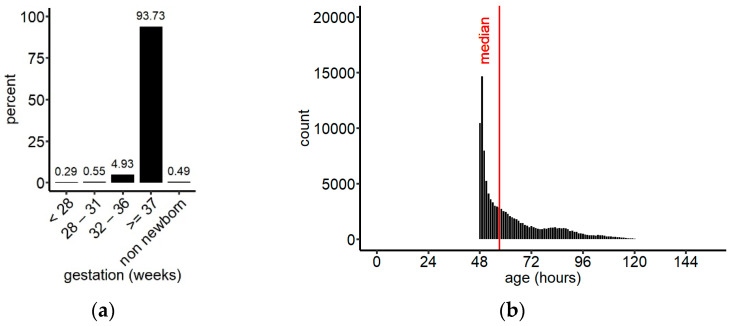
(**a**) Grouping of the screened population by gestational age; (**b**) distribution of sampling ages for newborns irrespective of their gestational age, binned in 1 h intervals. Median (IQR) = 57.1 h (49.9, 72.9); overall, 174 samples were collected after 150 h (data not shown). *n* = 115,211.

**Figure 3 IJNS-07-00059-f003:**
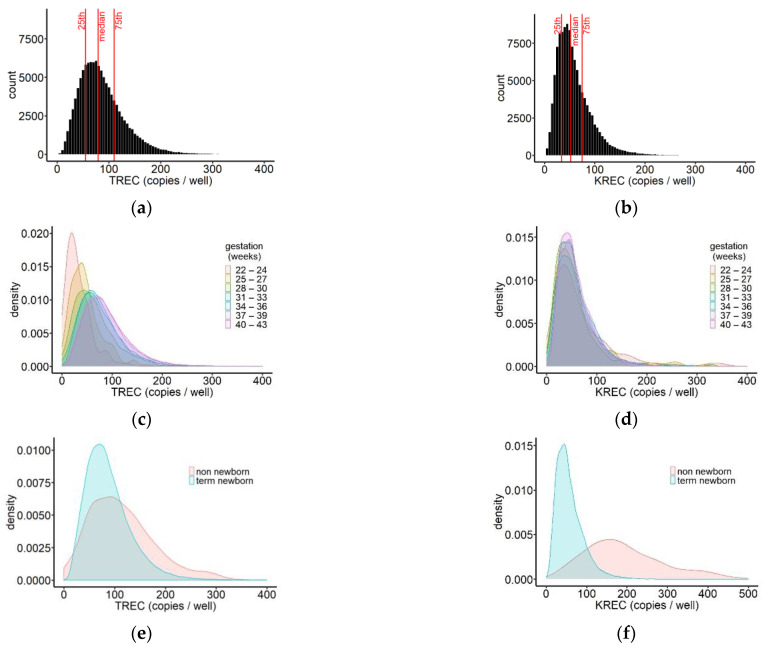
Distribution of TREC (**a**) and KREC (**b**) values in the screened population. Median (IQR)_TREC_ = 79 (55, 110); overall, 55 samples had TREC values above 400 copies/well (data not shown); median (IQR) _KREC_ = 51 (34, 75), and 42 samples had KREC values above 400 copies/well (data not shown); *n* = 115,757. Distribution of TREC (**c**) and KREC (**d**) values in newborns for different gestational ages represented by density plots. *n* = 115,189; samples above 400 copies/well are not shown; (**e**) density plot of the distribution of TREC values in screened non-newborns (*n* = 563), compared to term newborns (*n* = 108,508), showing a comparatively larger percentage of non-newborn children with very low or very high TREC concentrations. The median TREC concentration was 103 copies/well (IQR 64, 150) in non-newborns and 80 copies/well (IQR 56, 111) in term newborns; (**f**) density plot of the distribution of KREC values in screened non-newborns and term newborns showing a right shift of the KREC results for non-newborns in comparison to term newborns. The median KREC concentration was 180 copies/well (IQR 135, 258) in non-newborns and 51 copies/well (IQR 34, 74) in term newborns.

**Figure 4 IJNS-07-00059-f004:**
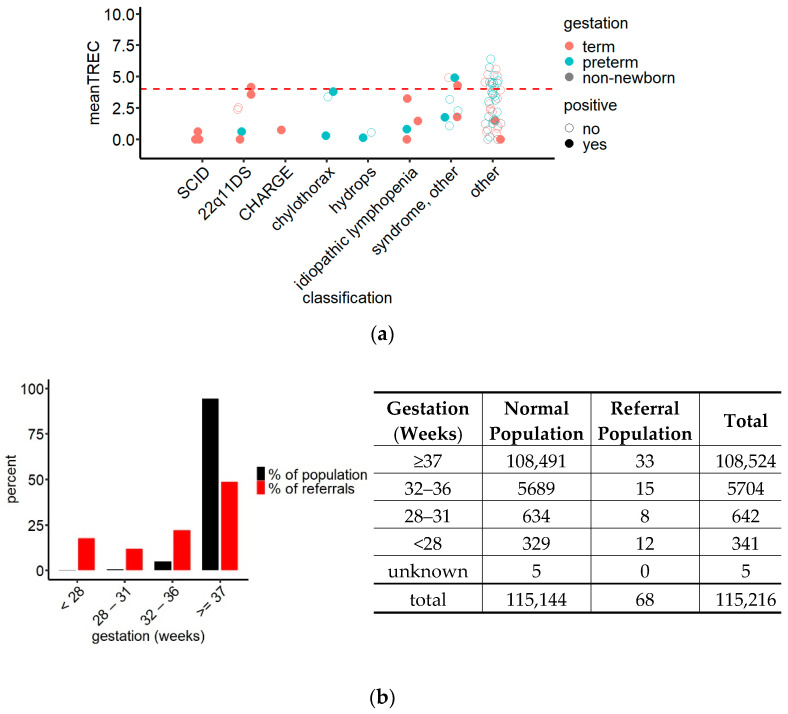
(**a**) TREC values of referrals grouped by clinical diagnosis and gestational age demonstrating the accumulation of the TREC values of TP (SCID and non-SCID lymphopenia, filled circles) in the lower half of the referral interval while the TREC values of FP (non-lymphopenic children, empty circles) were spread throughout the entire referral interval. Red dashed line = hypothetical referral cut-off at TREC 4 copies/well. *N* = 73; (**b**) comparison of the contribution of gestational age groups to the screened newborn population and to the referrals (left) and crosstabulation of the data underlying the statistical analysis (right) revealing an overrepresentation of preterm children in the referred population (red) compared to their frequency in the screened population (black); chi-squared test, x^2^ = 900.75, df = 4, *p* < 0.0001.

**Table 1 IJNS-07-00059-t001:** Median TREC and KREC values in children of different gestational ages.

Gestation (Weeks)	*n*	TREC (Copies/Well) Median (IQR)	KREC (Copies/Well) Median (IQR)
22–24	109	25 (14, 47)	49 (29, 85)
25–27	232	41 (25, 60)	47 (29, 74)
28–30	430	56 (36, 86)	46 (28, 66)
31–33	1 088	67 (44, 96)	50 (31, 77)
34–36	4 819	69 (47, 98)	50 (32, 73)
37–39	52 839	78 (55, 108)	53 (35, 78)
40–43	55 672	81 (56, 112)	49 (33, 72)

**Table 2 IJNS-07-00059-t002:** Clinical performance of the screening for the outcomes SCID and T cell lymphopenia. In total, 73 of 115,786 children were referred due to screen positive samples; see also Figure 1.

	SCID	T Cell Lymphopenia
Number of samples	115,786	
Number of screening negative samples	115,704	
Number of screening positive samples	73	
Number of true positives	3	21 ^1^
Number of false positives	70	52
% Sensitivity (95% CI)	100 (43.85–100)	-
% Specificity (95% CI)	99.937 (99.92–99.95)	99.953 (99.94–99.97)
% PPV	4.11 (1.41–11.40)	28.77 (19.65–40.01)
% NPV	100 (100–100)	-

^1^ Including the 3 SCID cases.

**Table 3 IJNS-07-00059-t003:** Average TREC and KREC results of the first and second screening cards and the clinical diagnosis for true positive cases. TREC and KREC concentrations in copies/well.

Case	1st Guthrie Card	2nd Guthrie Card	Total Lymphocyte Count (10^9^/L)	CD3 Count (10^9^/L)	Diagnosis
1	T 0; K 125	T 0; K 120	0.85	0.14	SCID, JAK3 deficiency
2	T 1; K 1	T 1; K 0	0.07	0.04	SCID, ADA deficiency
3	T 0; K 140	T 0; K 210	1.0	<0.01	SCID, JAK3 deficiency
4	T 0; K 65	not sampled	1.63	0.63	22q11DS
5	T 4; K 44	T 8; K 32	1.95	1.29	22q11DS
6	T 4; K 64	T 5; K 150	2.8	1.2	22q11DS
7	T 1; K 65	T 24; K 140	3.8	1.9	22q11DS, premature
8	T 1; K 52	T 1; K 55	1.7	1.14	CHARGE syndrome
9	T 4; K 8	T 1; K 18	0.2	0.32	chylothorax, premature
10	T 0; K 8	T 3; K 39	0.84	0.29	chylothorax, premature
11	T 0; K 54	T 0; K 21	1.3	0.09	hydrops, premature
12	T 4; K 110	T 180; K 98	0.45	0.33	idiopathic lymphopenia
13	T 1; K 260 ^1^	T 6; K 380	5.5	1.81	idiopathic lymphopenia
14	T 0; K 150	T 0; K 52	1.8	0.5	idiopathic lymphopenia
15	T 1; K 88	T 6; K 280	1.8	1.18	idiopathic lymphopenia, premature
16	T 4; K 36	T 7; K 70	1.84	1.2	syndrome ^2^
17	T 2; K 16	T 2; K 23	3.0	0.62	syndrome ^2^
18	T 6; K 8	T 2; K 95	1.4	0.3	syndrome, chylothorax, premature ^2^
19	T 2; K 15	deceased	not tested	not tested	syndrome, premature ^2^
20	T 2; K 27	T 12; K 74	2.0	0.61	other, sepsis
21	T 0; K 660	T 0; K 1000	10.1	1.88	other, juvenile myelomonocytic leukaemia

^1^ TREC = 1.45, rounded to 1; ^2^ syndromes other than 22q11DS or CHARGE.

## Data Availability

The data are not publicly available due to ethical reasons.

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
