# Peer review of "First Year of TREC-Based National SCID Screening in Sweden"

_2409-515X, 2021, doi:10.3390/ijns7030059_

Round 1

Reviewer 1 Report

In this manuscript, the authors describe the first year of newborn screening for SCID in Sweden. They describe their use of a multiplex assay for TREC, KREC, and ACTB quantitation and the results from the nearly 116,000 samples that were tested. Of particular interest is their reporting on TREC and KREC values from non-newborn children up to two years of age. Clarification of a few points would enhance the manuscript:

Figure 1 – In the middle column of the diagram, does there need to be a “Ø” preceding “TREC” in the top two boxes? Also, for replicates with TREC > 6, did the ACTB value need to be ≥ 1000 for the sample to screen negative? What was the outcome for samples with TREC > 6 and ACTB < 1000? If all replicates with TREC > 6 were deemed to screen negative, then the top box in the middle column should not have the “replicates with ACTB ≥ 1000” qualification.

Line 268 – Please indicate how many of the 16 newborns with undetectable TRECs were preterm.

Lines 281-283 – For the three children who had had second screening cards collected because of TPN treatment, how did this preclude collection of an additional sample on the day of examination or testing of the already-collected second screening card for SCID? Please clarify.

Figure 4 – If empty circles represent false positives, how are there any in any classification besides “other”? For example, if a child is diagnosed with 22q11DS, how can their screen also be a false positive? Please explain.

Reviewer 2 Report

The manuscript of Göngrich et al. describes the results of first year of newborn screening for SCID in the Sweden. 115 786 22 newborns were screened for SCID with a commercially available TREC/KREC/ACTB multiplex assay from August 2019 to August 2020 resulting in the identification of three SCID patients with mutations in JAK3 (N=2) and ADA (N=1). Other cases of T-cell lymphopenia (N=18) defined as secondary outcomes and false-positive referrals are defined as well. The authors describe significantly lower TREC levels in preterm infants, but remarkably did not observe any difference in KREC values between preterm and term born infants. Interestingly, TREC-levels were also reported for non-newborns up to the age of 2 years.

Since NBS for SCID was first introduced more than a decade ago, the novelty of the manuscript is somewhat reserved. However, it is of great importance that different NBS programs keep sharing their screening algorithms, screening results and clinical outcomes to enable international shared learning and to allow optimization of current NBS programs for SCID. For this reason, this manuscript is relevant to all NBS programs that are considering implementation of NBS for SCID or that are currently screening for SCID. It is therefore of high interests to the readers of IJNS. The authors have put forward some interesting new aspects of NBS for SCID including KREC-data and TREC/KREC data of non-newborns. The paper is clearly written, concise and thoughtfully discussed.

Some remarks:

  • Line 54. In more recent publications, survival with active infection at time of HSCT is quite improved: https://pubmed.ncbi.nlm.nih.gov/29021228/. I would consider using a more recent publication.
  • Line 124. The authors state that in case of extreme variability in the PCR results, outliers were removed from the calculation. Could the authors elaborate on this? What are considered outliers? If higher TREC levels are measured but f.e. ACTB would also be higher, would excluding these results not result in more false positive referrals? If a newborn has f.e. TRECs of 3, 6, 6 and 17 copies/well, I would state that this is not a SCID patient, but if you remove the outlier, the newborn would be referred.
  • Line 139. I just want to express my compliments to the authors for the clear definitions of false-positives in their manuscript.
  • Figure 3f. KREC levels were high in non-newborns compared to newborns. Is they any underlying immunological hypothesis that the authors could think of to explain these findings?
  • Line 260. There were eight newborns lost to follow-up before a second DBS card could be obtained. This seems like a relatively large proportion of the group? Could the authors elaborate on the reasons why these newborns were lost to follow-up? Were these newborns deceased? Is this observed for other disorders in the NBS program as well?
  • Line 267. The authors describe undetectable TRECs as TREC<1 copies/well, however they also report that the limit of detection of their TREC-assay is 3.41 copies/well. Is it therefore not true that no distinction can be made between TREC 0 to 3.41 copies/well? Could the authors comment?
  • Line 278. The median age of flow cytometry for preterms after an abnormal TREC result is 47.8 days. There is quite some time between referral and diagnostics for this group. I understand that T-cell populations will most likely normalize with time, but could the authors elaborate why they have not chosen to repeat a NBS sample in this time frame? If TRECs have normalized in a second NBS card, additional diagnostics would not have been required, cost could be saved.
  • Line 304: Can the authors explain the high TREC values measured in the case with a significant T-cell lymphopenia? I have never
  • Line 480. Were no IL2RG SCID patients clinically diagnosed in the period between the pilot study and national implementation of SCID screening in Sweden?

Some minor comments:

  • Line 37. of B-cell dysfunction…. Should this be due to?
  • Line 39. Reference 2. The most recent IUIS update is in 2019: DOI: 10.1007/s10875-019-00737-x
  • Line 310-312: a lot of overlap with the previous paragraph, I would consider removing the overlap.

Reviewer 3 Report

The manuscript by Christina Gongrich et al describes the findings of the first year of TREC based newborn screening for SCID in Sweden. The authors compiled comprehensive data on >100,000 newborns and children up to 2 years of age. Both TREC and KREC screening was performed on this group. The authors describe refinement of the cut-off values for TREC to decrease the number of false positives. They also note the challenges of developing TREC cut-offs for premature infants who often have false positive SCID newborn screens. The authors note that the TREC assay also picks up non-SCID T cell lymphopenic disorders. Overall, this manuscript is well written, informative, and contributes to the global data on SCID newborn screening. 

A minor comment for the authors:

Figure 1 legend: Include a note stating that the referral cut-off is =<6 copies/well. This information appears later in the text, after the figure and is confusing as the text directly above it mentions the referral cut-off, but does not state what the criteria are. 
